# Maternal Diabetes Mellitus and Neonatal Outcomes in Bisha: A Retrospective Cohort Study

**DOI:** 10.3390/medsci12020021

**Published:** 2024-04-15

**Authors:** Abdullah Alshomrany, Elhadi Miskeen, Jaber Alfaifi, Hassan Alshamrani, Abdulmohsen Alshahrani

**Affiliations:** 1College of Medicine, University of Bisha, Bisha 67714, Saudi Arabia; hassanshamranial@gmail.com (H.A.); abdulmohsenalsh996@gmail.com (A.A.); 2Department of Obstetrics and Gynaecology, College of Medicine, University of Bisha, Bisha 67714, Saudi Arabia; emiskeen@ub.edu.sa; 3Department of Pediatrics and Child Health, College of Medicine, University of Bisha, Bisha 67714, Saudi Arabia; jalfaifi@ub.edu.sa

**Keywords:** maternal diabetes mellitus, type 1 diabetes mellitus, type 2 diabetes mellitus, gestational diabetes mellitus, neonatal outcomes

## Abstract

Background: Maternal diabetes mellitus (MDM) is associated with increased risks for adverse neonatal outcomes. However, the impact of MDM on neonatal outcomes in Bisha, a city in Saudi Arabia, is not well documented. This study aims to investigate the impact of MDM on neonatal outcomes in the Maternity and Children’s Hospital (MCH), Bisha, Saudi Arabia. Methods: A retrospective cohort study was conducted on 181 pregnant women with diabetes and their neonates who were diagnosed at the Maternity and Children’s Hospital (MCH), Bisha, Saudi Arabia, between 5 October 2020 and 5 November 2022. The primary outcome was a composite of adverse neonatal outcomes, including stillbirth, neonatal death, macrosomia, preterm birth, respiratory distress syndrome, hypoglycemia, and congenital anomalies. Logistic regression analyses were used to adjust for potential confounders. Results: The total sample size was 181. The average age of patients was 34 years (SD = 6.45). The majority of the patients were diagnosed with GDM, 147 (81.2%), and pre-GDM, 34 (18.8%). Neonates born to mothers with MDM had a higher risk of adverse neonatal outcomes compared to those born to mothers without MDM (adjusted odds ratio [aOR] = 1.46, 95% confidence interval [CI]: 1.25–1.70). The risks of macrosomia (aOR = 1.74, 95% CI: 1.38–2.19), LBW (aOR = 1.32, 95% CI: 1.06–1.66), and RDS (aOR = 1.57, 95% CI: 1.28–1.93) were significantly higher among neonates born to mothers with MDM. The types of DM were statistically significant in terms of their correlation with the following neonatal outcomes: hypoglycemia (*p*-value = 0.017), macrosomia (*p*-value = 0.050), and neonatal death (*p*-value = 0.017). Conclusions: MDM is associated with an increased risk of adverse neonatal outcomes in Bisha. The early identification and management of MDM may improve neonatal outcomes and reduce the burden of neonatal morbidity and mortality in this population.

## 1. Introduction

Diabetes mellitus (DM) is a metabolic disorder characterized by chronic hyperglycemia or elevated blood glucose levels beyond the normal range [1]. During pregnancy, diabetes is classified as pregestational (pre-GDM) or gestational diabetes (GDM). Pre-GDM occurs when women are diagnosed with type 1 or type 2 diabetes mellitus before pregnancy. Type 1 diabetes mellitus (T1DM) is caused by an autoimmune reaction leading to the destruction of β cells in the pancreas, resulting in insulin deficiency, whereas type 2 diabetes mellitus (T2DM) is caused by inadequate insulin production from β cells in the pancreas and insulin resistance in peripheral tissues [2,3]. GDM is a carbohydrate metabolism impairment or carbohydrate intolerance that is often diagnosed in the second or third trimester of pregnancy and is strongly associated with type 2 diabetes after pregnancy [2,3,4].

According to the International Diabetes Federation (IDF), the worldwide incidence of diabetes in adults aged 20–79 was 10.5% in 2021 and is projected to increase to 11.3% by 2030 [1]. In Saudi Arabia, diabetes mellitus affects 30% of the population [5], while the prevalence of diabetes in pregnancy was found to be approximately 16.7% in 2021 [1]. Globally, the prevalence of diabetes during pregnancy is on the rise, as is the prevalence of diabetes in the general population. Maternal diabetes mellitus affects approximately one out of every six live birth cases; 85% are GDM, and 15% are pre-GDM [6]. In Saudi Arabia, the prevalence of GDM is higher than in other countries, affecting 19.6% of pregnant women [7]. In Riyadh, the prevalence of diabetes among all pregnancies ranges from 4.3% to 24.3%, including pre-GDM and GDM. Out of 9723 women, 24.2% were found to have GDM and 4.3% had pre-GDM [8]. GDM’s prevalence varies widely due to obesity, diabetes epidemics, and advanced maternal age during pregnancy, and these conditions continue to worsen globally [4].

Pregnant women with well-controlled diabetes through a healthy diet, exercise, and appropriate body weight generally have healthy neonates. However, uncontrolled maternal diabetes is strongly associated with cesarean deliveries and operative vaginal deliveries [6,9]. Uncontrolled maternal diabetes can adversely affect neonatal health, leading to metabolic and hematologic disorders, respiratory distress, cardiac disorders, and neurologic impairment due to perinatal asphyxia and birth trauma [10]. Both pre-GDM and GDM are strongly linked to unfavorable pregnancy outcomes [11]. Diabetes during pregnancy is associated with significant short- and long-term effects, such as an increased risk of obesity and diabetes development in both mothers and children, as well as extremely high healthcare costs [4,12].

To date, no research has been conducted on the impact of maternal diabetes on neonatal outcomes in Bisha. Therefore, investigating the effects of maternal diabetes on neonatal health in Bisha, Saudi Arabia, is essential. This study aims to determine the impact of maternal diabetes on neonatal outcomes at the Maternity and Children’s Hospital (MCH) in Bisha.

## 2. Materials and Methods

This study utilized a retrospective cohort design to examine 181 pregnant women with diabetes and their neonates who were diagnosed at the Maternity and Children’s Hospital (MCH) in Bisha, Saudi Arabia. The study took place in the Bisha province in southwestern Saudi Arabia, which has a population of approximately 398,256. The Maternity and Children’s Hospital (MCH), Bisha, was the primary location for the study, with a bed capacity of 100 and 41 groups.

The study population included all pregnant women with diabetes and their neonates who had complete medical records between 5 October 2020 and 5 November 2022. The study excluded women under 18, those planning to give birth at a different hospital, those without diabetes, those who were not currently pregnant, and those with incomplete medical records. Patients either suffering from comorbidities or administered concomitant treatments were excluded.

After obtaining permission from the MCH, Bisha, the study collected antenatal, perinatal, and postnatal data from patient medical records, including demographic data and clinical information on pregnancy and delivery characteristics. Data on neonatal morbidity and mortality, including birth weight, respiratory distress syndrome (RDS), low birth weight (LBW), neonatal hypoglycemia, neonatal death, admission to the neonatal intensive care unit (NICU), cardiac disorders, neurologic impairment due to perinatal asphyxia, and birth trauma were also collected. Each neonate’s status was followed up after birth.

Data were entered into Microsoft Office Excel 2019 and analyzed using the Statistical Package for Social Science (SPSS) version 23. The analysis involved providing a complete description of the dataset using the numbers, frequencies, and percentages of the variables in the study. Bivariate analysis or cross-tab procedures were used to test the dependent variables against each predictor variable. Any *p*-value of ≤0.05 was considered significant.

The study defined T1DM, T2DM, GDM, neonatal macrosomia, and LBW. The study received ethical clearance from the University of Bisha College of Medicine (UBCOM) ethical committee with the registration number H-06-BH-087, and permission from the MCH, Bisha, was obtained. Informed consent was obtained from all subjects involved in the study.

## 3. Results

The Maternity and Children’s Hospital (MCH) in Bisha, Saudi Arabia, was the site of this study involving 181 pregnant women with diabetes and their neonates. The study revealed a connection between maternal diabetes mellitus and unfavorable neonatal outcomes in Bisha. These results underscore the significance of the timely diagnosis and effective management of diabetes during pregnancy as a means of enhancing neonatal outcomes in the region.

### 3.1. Maternal Age

The average patient age was around 34 years (SD = 6.45), as shown in Table 1.

### 3.2. Maternal Characteristics

Table 2 presents the maternal characteristics of the participants: The majority of the patients were Saudi, 168 (92.8%), and from urban areas, 148 (81.8%). Most of the patients had no family history of diabetes, 141 (77.9%). The majority of the patients were diagnosed with diabetes mellitus during pregnancy (GDM), 147 (81.2%). Most of the patients had three babies or more, 115 (63.5%). Most of the patients were delivered by C-section, 172 (95%). The patients had a history of neonatal death, 12 (6.6%). Birth weight was found to be less than 2.5 kg (low birth weight) in 21 (11.6%) neonates and more than 4 kg (macrosomia) in 32 (17.7%) neonates. The types of diabetes were statistically significant in terms of their correlation with the maternal characteristics time-diagnosed DM (*p*-value = 0.010) and type of delivery (*p*-value = 0.049), as shown in Table 2.

### 3.3. Neonatal Outcomes

Table 3 presents the neonatal outcomes. The majority of the neonates presented with a normal condition, 87 (48.1%); congenital heart disease, 39 (21.5%); macrosomia, 28 (15.5%); LBW, 21 (11.6%); RDS, 16 (8.8%); sepsis, 12 (6.6%); DM, 3 (1.7%); development and growth disorder, 2 (1.1%); hypoglycemia, 1 (0.6%); prematurity, 1 (0.6%); and neonatal death, 1 (0.6%). Most neonates after birth were cured, 106 (59.6%), or experienced an improvement in their condition, 66 (36.5%), while some neonates developed complications, 8 (4.4%), and some died, 1 (0.6%). The frequency of NICU admission among neonates was 65 (35.9%). The types of diabetes were statistically significant in terms of their correlation with the neonatal outcomes hypoglycemia (*p*-value = 0.017), macrosomia (*p*-value = 0.050), and neonatal death (*p*-value = 0.017), as shown in Table 3.

### 3.4. Diagnosis of DM in Pregnancy

The frequencies and percentages of the types of DM are described in Figure 1.

### 3.5. Logistic Regression

Neonates born to mothers with MDM had a higher risk of adverse neonatal outcomes compared to those born to mothers without MDM (adjusted odds ratio [aOR] = 1.46, 95% confidence interval [CI]: 1.25–1.70). The risks of macrosomia (aOR = 1.74, 95% CI: 1.38–2.19), preterm birth (aOR = 1.32, 95% CI: 1.06–1.66), and respiratory distress syndrome (aOR = 1.57, 95% CI: 1.28–1.93) were significantly higher among neonates born to mothers with MDM, as shown in Table 4.

## 4. Discussion

This study describes the maternal characteristics and neonatal outcomes in 181 women. The main outcome of maternal diabetes is delivering neonates with surgical intervention. In the present study, most of the patients were delivered by CS in both GDM and pre-GDM mothers. The outcomes of this study are supported by data gathered from other regions of the nation.

In the present study, GDM was more common than pre-GDM, and these results are supported by studies conducted in Saudi Arabia (SA) and India, which both showed that GDM was more common than pre-GDM [8,13]. The present study attempted to categorize pregestational diabetes into T1DM and T2DM. The frequency of pregestational T2DM was higher than that of T1DM, which agrees with previous reports [11] but differs from studies conducted in Scotland and Ireland, which showed that T1DM is more common than T2DM [14,15]. Due to the continuous elevation in GDM incidence resulting from the ongoing enrichment over time of risk factors of GDM, such as obesity, advanced-age pregnancy, and the increased number of births [16], patients with pre-GDM or GDM should receive more attention from their healthcare providers to prevent the development of unfavorable outcomes.

In GDM, unexpected outcomes that differed from those of previous studies were obtained in this research, describing that almost half of the patients were delivered by CS and the other half were delivered vaginally, but this differs from a study conducted in Saudi Arabia in which vaginal delivery was more common than CS [8,17,18,19]. In pre-GDM, unexpectedly, different outcomes were obtained between this study and previous studies describing vaginal delivery as more common than CS [8,20]. It was also noted that that not all patients who undergo CS should be viewed as having unfavorable pregnancy outcomes; instead, CS is frequently suggested as a preventive measure used by healthcare professionals to reduce the risk of perinatal problems brought on by maternal diabetes [21]. So, in maternal diabetes, healthcare providers should use strategies and recommendations to prevent unfavorable outcomes.

Furthermore, in the present study, macrosomia was a more common adverse outcome associated with maternal diabetes than LBW. Additionally, macrosomia in GDM was more common than in LBW, and these results are supported by studies conducted in China and Qatar [22,23]. Both studies found macrosomia to be more common than LBW, but these results are different from those of studies conducted in Saudi Arabia (SA) and Brazil that showed that low birth weight in GDM is more common than macrosomia [18,24]. In the present study, in pre-GDM, LBW is more common than macrosomia, which was expected, but this differs from the results of a study conducted in Italy that showed a high association between macrosomia and pre-GDM [11]. As expected, women with all types of maternal diabetes were at risk of developing macrosomia or LBW. So, patients with pre-GDM or GDM should be careful to maintain glycemic control and attend regular follow-ups, and quick treatment should be provided for diabetic mothers to prevent the development of macrosomia or LBW.

The present study evaluated different neonatal outcomes according to the types of maternal DM. The major neonatal outcomes of maternal DM included congenital heart disease, macrosomia, RDS, and NICU admission. The findings of this study are supported by data gathered from other studies conducted in Saudia Arabia. In the present study, in maternal diabetes, the most common neonatal finding was congenital heart disease; these results are supported by studies conducted in New York that showed that congenital heart disease had a strong association with maternal diabetes, and pre-GDM had a more significant association with congenital heart disease phenotypes and categories [25,26]. In this study, macrosomia was the second most common neonatal finding related to GDM, which is supported by studies showing that GDM is associated with macrosomia [25,27]. Moreover, RDS was strongly associated with all types of maternal DM, which is supported by studies showing that GDM and pre-GDM has a greater association with RDS [26,28,29,30], and Sepsis is related to maternal diabetes, which is also supported by a study that showed a stronger relationship between sepsis and maternal DM and hypertension [31]. In this research, an association between maternal diabetes and neonatal death was found, which is supported by previous studies that suggested an association with neonatal mortality [32], but is different from a study that showed no association between maternal diabetes and neonatal necrotizing enterocolitis, intraventricular hemorrhage, or neonatal death [33]. In the present study, admission to the NICU was associated with GDM and pre-GDM, which is supported by studies showing a relationship between GDM and pre-GDM and NICU admissions [18,33]. We assert that during intrapartum care for women affected by diabetes, closely monitoring glucose levels during labor may contribute to reducing the occurrence of postnatal hypoglycemia in newborns as well as neonatal morbidities. However, these results may agree or disagree with those of other studies because of variations in the patient groups investigated and advancements in neonatal care at delivery. Herewith, we highlight the findings of a recent study suggesting that the risk of neonatal hypoglycemia appears to be similar between offspring exposed to single metformin treatment and those undergoing nutrition therapy alone [34].

The association between polycystic ovary syndrome (PCOS) and gestational diabetes (GD) is an area of growing interest and research. Nutraceutical supplementation with inositols, alpha-lipoic acid, vitamin D, and metformin may hold promise in reducing the risk of GD in PCOS patients, although further research is needed to confirm their efficacy. We can benefit from the growing literature addressing clinical improvement of patient care [35].

Currently, in Saudi Arabia, the prevalence of maternal diabetes is around 28.5% of all pregnancies [8]. With the growth of the public health burden of GDM and pre-GDM, clinical management must be informed by high-quality research to achieve the best possible outcomes for neonates. In this retrospective cohort, compared to GDM or no diabetes, maternal pre-GDM was linked to an increase in severe neonatal morbidity. In contrast, neonates born to mothers with pre-GDM are most at risk of severe neonatal morbidity and may still experience unfavorable outcomes, needing the greatest care throughout delivery.

The weakness of our study included its retrospective approach, which, while helpful in providing insights into past data, can be limited in terms of its ability to capture all relevant information. Additionally, the reliance on administrative data may not provide a complete picture of the outcomes for both mothers and neonates. The lack of information on maternal glycemic control is also a limitation as it can impact both maternal and neonatal outcomes. Without these data, it is challenging to draw any definitive conclusions on the relationship between maternal glycemic control and outcomes. Additionally, patients did not perform an Oral Glucose Tolerance Test following delivery. Overall, it is important to consider these limitations when interpreting the results of this study and to be cautious in drawing conclusions based solely on its findings.

The strengths of our study included identifying important associations between GDM and pre-GDM patients with increased risk of neonatal mortality and morbidity. This provides valuable information for healthcare providers and policymakers to improve prenatal care and the management of diabetes in pregnant women. Additionally, this study’s large sample size and use of national administrative data provide a broad perspective on the issue and increase the generalizability of the findings.

The implication of our results in practice is that healthcare providers should consider conducting a prospective study to evaluate diabetic women from their first visit to the time of delivery. This type of study would allow for a more comprehensive evaluation of maternal glycemic control and its relationship with both maternal and neonatal outcomes. Additionally, it would provide more accurate data on these outcomes and could potentially identify interventions that could improve outcomes for mothers with GDM or pre-GDM and their neonates. It is also important for healthcare providers to provide education and support to women with GDM or pre-GDM to help them manage their condition and minimize the risks associated with it.

While this study provides valuable insights into the association between MDM and neonatal outcomes, its retrospective nature and certain limitations underscore the need for further research to validate its findings and address potential confounders. Nonetheless, this study’s strengths contribute to the growing body of evidence informing clinical practice in managing MDM and improving neonatal health outcomes.

## 5. Conclusions and Recommendations

It is crucial for healthcare providers to identify and manage DM in pregnant women, as this can lead to improved maternal and neonatal outcomes. The findings of this study highlight the importance of closely monitoring and managing blood glucose levels in pregnant women with DM, in order to reduce the risk of adverse neonatal health outcomes such as macrosomia, NICU admission, LBW, and RDS. This study also emphasizes the need for further research to better understand the relationship between glycemic control and maternal and neonatal outcomes in women with DM during pregnancy. Overall, these findings have important implications for clinical practice and can inform the development of guidelines for the management of DM in pregnant women.

The above is a very comprehensive and informative statement. Providing early and successful interventions, such as prenatal care and careful glycemic control, can greatly improve the outcomes for both the mother and the neonate. Screening every pregnant woman for pre-GDM and GDM can help identify those at risk early on and enable efforts to prevent their emergence. Educating pregnant women on the importance of maintaining adequate glycemic control and preventing the development of type 2 diabetes in GDM patients can also help reduce the impact of maternal diabetes on society. Overall, a multidisciplinary approach involving obstetricians, endocrinologists, and other healthcare professionals is necessary to improve outcomes in pregnant women with diabetes.

## Figures and Tables

**Figure 1 medsci-12-00021-f001:**
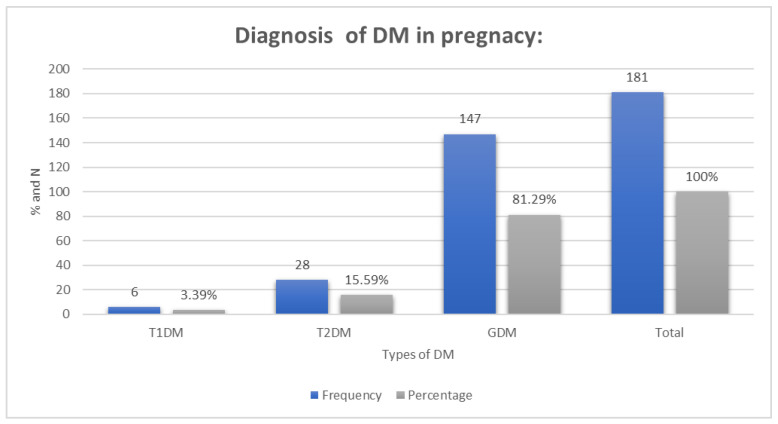
Clustered bar chart showing the frequencies and percentages of the types of DM.

**Table 1 medsci-12-00021-t001:** Pregnant women’s ages and their frequencies.

Age Group:	Frequency	Percent
21–25	21	11.7%
26–30	25	13.8%
31–35	48	26.5%
36–40	51	28%
41–46	36	20%
Total	181	100%

**Table 2 medsci-12-00021-t002:** Maternal characteristics of the three groups.

Maternal Characteristics		T1DM	T2DM	GDM	Total	*p*-Value
Nationality	Saudi	6 (3.24%)	26 (14.35%)	136 (75.12%)	168 (92.8%)	0.056
	Non-Saudi	0 (0%)	2 (1.30%)	11 (6.09%)	13 (7.2%)	
Region	City (urban)	5 (2.76%)	24 (13.26%)	119 (65.77%)	148 (81.8%)	
	Village (rural)	1 (0.55%)	1 (0.55%)	24 (13.29%)	26 (14.4%)	0.076
	Outside the region	0 (0%)	3 (%)	4 (%)	7 (3.9%)	
Family history of diabetes	Yes	0 (%)	7 (%)	33 (%)	40 (22.1%)	0.061
	No	6 (%)	21 (%)	114 (%)	141 (77.9%)	
Time diagnosed with DM	Before pregnancy (pre-GDM)	6 (%)	28 (%)	0 (%)	34 (18.8%)	0.010 *
	During pregnancy (GDM)	0 (0%)	0 (0%)	147 (81.2%)	147 (81.2%)	
Number of pregnancies	1–2	4 (2.2%)	6 (3.31%)	56 (30.96%)	66 (36.5%)	
	3–4	2 (1.1%)	13 (7.18%)	61 (33.7%)	76 (42%)	0.074
	More than 5	0 (0%)	9 (4.96%)	30 (16.5%)	39 (21.5%)	
Type of delivery	Vaginal	0 (0%)	1 (0.55%)	8 (4.44%)	9 (5%)	0.049 *
	CS	6 (3.3%)	27 (14.9%)	139 (76.77%)	172 (95%)	
History of neonatal death	Yes	1 (0.55%)	3 (1.65%)	8 (4.4%)	12 (6.6%)	0.096
	No	5 (2.76%)	25 (13.81%)	139 (76.82%)	169 (93.4%)	
Birth weight	<2.5 LBW	5 (2.7%)	5 (2.6%)	11 (6.07%)	21 (11.6%)	
	2.5–3.99 normal	1 (0.55%)	20 (11.04%)	107 (59.10%)	128 (70.7%)	0.078
	≥4 kg (macrosomia)	0 (0%)	3 (1.65%)	29 (%)	32 (17.7%)	

Type 1 diabetes mellitus (T1DM); type 2 diabetes mellitus (T2DM); gestational diabetes malleus (GDM); cesarean section (CS); low birth weight (LBW); * indicate significant *p*-value.

**Table 3 medsci-12-00021-t003:** Neonatal outcomes of the three groups.

Neonatal Outcome		T1DM	T2DM	GDM	Total	*p*-Value
RDS		2 (1.1%)	3 (1.65%)	11 (6%)	16 (8.8%)	0.102
Prematurity		1 (0.6%)	0 (0%)	0 (0%)	1 (0.6%)	0.128
Hypoglycemia		0 (0%)	0 (0%)	1 (0.6%)	1 (0.6%)	0.017 *
Congenital heart disease		0 (0%)	13 (7.16%)	26 (14.3%)	39 (21.5%)	0.076
Development and growth disorder		0 (0%)	2 (1.1%)	0 (0%)	2 (1.1%)	0.062
DM		0 (0%)	1 (0.5%)	2 (1.1%)	3 (1.7%)	0.073
LBW		5 (2.7%)	5 (2.7%)	11 (6.07%)	21 (11.6%)	0.106
More than 4 kg (macrosomia)		0 (0%)	3 (1.66%)	25 (13.8%)	28 (15.5%)	0.050 *
Sepsis		2 (1.1%)	1 (0.55%)	9 (4.95%)	12 (6.6%)	0.111
Neonatal death		0 (0%)	0 (0%)	1 (0.6%)	1 (0.6%)	0.017 *
Normal		1 (0.5%)	9 (4.9%)	77 (42.5%)	87 (48.06%)	0.066
Neonatal outcomes	Cured	3 (1.68%)	11 (6.18%)	92 (51.7%)	106 (59.6%)	
	Died	0 (0%)	0 (0%)	1 (0.6%)	1 (0.6%)	0.080
	Developed complications and then improved	2 (1.1%)	15 (8.29%)	49 (27%)	66 (36.5%)	
	Developed complications	1 (0.55%)	2 (1.1%)	5 (2.75%)	8 (4.4%)	
NICU admission		3 (1.65%)	13 (7.18%)	49 (27%)	65 (35.9%)	0.077

Type 1 diabetes mellitus (T1DM); type 2 diabetes mellitus (T2DM); gestational diabetes malleus (GDM); respiratory distress syndrome (RDS); diabetes mellitus (DM); low birth weight (LBW); neonatal intensive care unit (NICU); * indicate significant *p*-value.

**Table 4 medsci-12-00021-t004:** Neonatal outcomes in MDM mothers and their prediction by logistic regression.

Neonatal Outcomes in MDM Mothers:	Odd Ratio (OR), 95% Confidence Interval (CI)
Macrosomia	OR = 1.74, 95% CI: 1.38–2.19
LBW	OR = 1.32, 95% CI: 1.06–1.66
RDS	OR = 1.57, 95% CI: 1.28–1.93

Odds ratio (OR), 95% confidence interval (CI), low birth weight (LBW), respiratory distress syndrome (RDS).

## Data Availability

The data that support this study’s findings are available from the corresponding author, A. Alshomrany, upon reasonable request.

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
