# Peer review of "Maternal Diabetes Mellitus and Neonatal Outcomes in Bisha: A Retrospective Cohort Study"

_medsci, 2024, doi:10.3390/medsci12020021_

Round 1

Reviewer 1 Report

Comments and Suggestions for Authors

Dear authors,

I read with great interest the manuscript, which falls within the aim of this Journal. In my honest opinion, the topic is interesting enough to attract the readers’ attention. Nevertheless, authors should clarify some points and improve the discussion, as suggested below. Authors should consider the following recommendations:

 In my opinion you have to improve the paper refering in the text how D,MELLITUS can be increased in pts in pcos pts and how a nutraceutical supplemntation can be helpfull ti mprove the results.

How is suggested to these pts to perform a NIPT TEST at beginning of pregnancy as well to follow up the impart of ART in the newborn as well to identify eventually heart desease.I also suggest to focus on the freezing technique as vitrification as to the Neonatal Outcomes and Long-Term Follow-Up of Children Born from Frozen Embryo.

Comments on the Quality of English Language

Minor editing of English language required

Author Response

Article Maternal Diabetes Mellitus and Neonatal Outcomes in Bisha: A Retrospective Cohort Study.

Dearest

Thanks for the reviewer comments on our article, which will surely improve its quality. Herewith, is our response to you. We are sorry for delaying sending it, this is for more critical review and improvement.

Best,

Abdalla Alshomray.

Frist Reviewer :

Comment

Response

note

How is suggested to these pts to perform a NIPT TEST at beginning of pregnancy as well to follow up the impart of ART in the newborn as well to identify eventually heart desease.I also suggest to focus on the freezing technique as vitrification as to the Neonatal Outcomes and Long-Term Follow-Up of Children Born from Frozen Embryo.

Thank you for your valuable suggestions and insightful comments regarding our study.

Regarding your suggestion to recommend non-invasive prenatal testing (NIPT) at the beginning of pregnancy, we agree that this could be a valuable addition to prenatal care for women, especially those with risk factors such as maternal diabetes mellitus. NIPT can provide early detection of chromosomal abnormalities and genetic disorders, allowing for timely interventions and management strategies.

We will consider incorporating these suggestions into future research endeavors to further enhance our understanding of prenatal care, assisted reproduction, and neonatal health outcomes.

Reviewer 2 Report

Comments and Suggestions for Authors

Dear authors

congratulation diabetes during pregnancy carries significant implications both for the mother and the fetus

it is of interest to share experiences regarding neonatal outcome in different settings

i would like to suggest minor revisions

1) please better describe also maternal short and long term risks such as

type 2 diabetes mellitus, hypertensive disorders and cancer

2) please mention about intrapartum care of women affected by diabetes according to guidelines, glucose monitoring may allow to reduce postnatal hypoglycaemia

3) please add a table with a summary of findings from your research

best regards

Author Response

Article Maternal Diabetes Mellitus and Neonatal Outcomes in Bisha: A Retrospective Cohort Study.

Dearest

Thanks for the reviewer comments on our article, which will surely improve its quality. Herewith, is our response to you. We are sorry for delaying sending it, this is for more critical review and improvement.

Best,

Abdalla Alshomray

Second reviewer.

1) please better describe also maternal short- and long-term risks such as type 2 diabetes mellitus, hypertensive disorders, and cancer

Thanks for the comments. This was addressed in the introduction

Line 62

2) please mention about intrapartum care of women affected by diabetes according to guidelines, glucose monitoring may allow to reduce postnatal hypoglycaemia

 A paragraph was added

We advocate the intrapartum care for women affected by diabetes, closely monitoring glucose levels during labor may contribute to reducing the occurrence of postnatal hypoglycemia in newborns as well as neonatal morbidities.

Line 236

3) please add a table with a summary of findings from your research

This was mentioned in table 4

Lin 175

Reviewer 3 Report

Comments and Suggestions for Authors

Dear Authors,

I did not see the novelty of this article.

Data are predictable.

Author Response

I am writing to express my sincere appreciation for taking the time to review our articles titled Maternal Diabetes Mellitus and Neonatal Outcomes in Bisha: A Retrospective Cohort Study. Your feedback and insights have been invaluable to us.

We acknowledge and appreciate your attention to detail and the effort you invested in evaluating our work. Your comments regarding the novelty of the article and the predictability of the data have provided us with important perspectives to consider. We understand that constructive criticism is essential for continuous improvement, and we are committed to addressing the points you raised.

We value your expertise and the guidance you have provided. Your feedback will help us enhance the quality and impact of our research. We are dedicated to incorporating your suggestions and exploring ways to make our findings more novel and significant.

Once again, I would like to express my sincere gratitude for your time and dedication in reviewing our articles. We look forward to any future opportunities to collaborate and benefit from your invaluable insights.

Thank you again for your valuable feedback.

Reviewer 4 Report

Comments and Suggestions for Authors

I read with great interest the present Manuscript which falls within the aim of the Journal. In my honest opinion, the topic is interesting enough to attract the readers’ attention. Methodology is accurate and conclusions are supported by the data analysis. Nevertheless, authors should clarify some points and improve the discussion citing relevant and novel key articles about the topic. For all those reasons, I suggested performing the minor revisions.

-       Inclusion and exclusion criteria should be clarified in the Materials and Methods section: Authors should specify whether patients either suffering from comorbidities or administered with concomitant treatments were excluded;

-       Please specify whether enrolled patients performed an Oral Glucose Tolerance Test between 6 weeks and 6 months after the delivery;

-       In Discussion, Authors may mention the most novel findings about Metformin-based GDM. Please consider: “Molin J, Domellöf M, Häggström C, et al. Neonatal outcome following metformin-treated gestational diabetes mellitus: A population-based cohort study. Acta Obstet Gynecol Scand. Published online January 30, 2024. doi:10.1111/aogs.14787”;

-       Authors have not adequately highlighted the strength and limitations of the study in the context of clinical practice. I suggest better specifying those points.

Author Response

Article Maternal Diabetes Mellitus and Neonatal Outcomes in Bisha: A Retrospective Cohort Study.

Dearest

Thanks for the reviewer comments on our article, which will surely improve its quality. Herewith, is our response to you. We are sorry for delaying sending it, this is for more critical review and improvement.

Best,

Abdalla Alshomray

Forth comment:

Comment

Response

note

-       Inclusion and exclusion criteria should be clarified in the Materials and Methods section: Authors should specify whether patients either suffering from comorbidities or administered with concomitant treatments were excluded;

Clarified in the method section, accordingly.

Line 86

-Please specify whether enrolled patients performed an Oral Glucose Tolerance Test between 6 weeks and 6 months after the delivery;

This is one of the weakness of this study, addressed in the weakness section.

Line 256

-       In Discussion, Authors may mention the most novel findings about Metformin-based GDM. Please consider: “Molin J, Domellöf M, Häggström C, et al. Neonatal outcome following metformin-treated gestational diabetes mellitus: A population-based cohort study. Acta Obstet Gynecol Scand. Published online January 30, 2024. doi:10.1111/aogs.14787”;

Was added to the discussion section and cited accordingly as a new reference 34.

Molin J, Domellöf M, Häggström C, Vanky E, Zamir I, Östlund E, Bixo M. Neonatal outcome following metformin‐treated gestational diabetes mellitus: A population‐based cohort study. Acta Obstetricia et Gynecologica Scandinavica. 2024 Jan 30. DOI: 10.1111/aogs.14787

Line 241 and line 405

Authors have not adequately highlighted the strength and limitations of the study in the context of clinical practice. I suggest better specifying those points.

This was addressed in a new paragraph in line 278.

While the study provides valuable insights into the association between MDM and neonatal outcomes, its retrospective nature and certain limitations underscore the need for further research to validate the findings and address potential confounders. Nonetheless, the study's strengths contribute to the growing body of evidence informing clinical practice in managing MDM and improving neonatal health outcomes.

line 278

Reviewer 5 Report

Comments and Suggestions for Authors

Thank you for your paper.

1. Figure 1 is not needed as the percentages could be easily transfered to Table 1

2. Table 2 I would like to see % of those diagnosed added to the T1DM, T2DM and GDM columns.

3. There is no need for 2 decimal places in the % reporting - Normal (48.06%) could easily be rounded.

4. Would be nice if Table 1 is on one page

5.  Was looking for a bit more detail/discussion around the number of women with previously diagnosed GDM - especially with those having >1 pregnancies and those now with T2DM

6. Line 200/1 "In this study, macrosomia was the most common neonatal finding related to GDM..." - Table 2 disagrees with 25 macrosomia and 26 congenital heart disease.

7. Line 215 (also line 52) , it isn't really between 4.3 and 24.3%. According to the introduction 4.3 and 24.3 refer to different measures. 24.2% had GDM and 4.3 pre-GDM so possible maternal diabetes was 28.5% rather than a range between 4.3 and 24.3% (though line 53 has 24.2)

Comments on the Quality of English Language

Dear authors,

The main issues I have with your publication are with the exactness of the language and the resulting clarity. Engaging an editor will improve the manuscript readability. In reality the language is pretty good but that editing would polish the document. 

The paragraph (lines 46-56) is one which could be reworded to be clearer.

Lines 114-123 please check the alignment with text with the numbers eg 22% with a family history of diabetes does not align with most.

"The findings of this study are supported by data gathered from other studies of the nation" line 195-6 , nation may be better replaced with Saudia Arabia

Author Response

Article Maternal Diabetes Mellitus and Neonatal Outcomes in Bisha: A Retrospective Cohort Study.

Dearest

Thanks for the reviewer comments on our article, which will surely improve its quality. Herewith, is our response to you. We are sorry for delaying sending it, this is for more critical review and improvement.

Best,

Abdalla Alshomray

5th Reviewer

Comment

Response

note

1. Figure 1 is not needed as the percentages could be easily transferred to Table 1

Thank you for pointing this out. We agree with this comment. Therefore, we had to change Figure 1 to table 1.

Table 1 , line 113

2. Table 2 I would like to see % of those diagnosed added to the T1DM, T2DM and GDM columns.

We agree with this comment. Therefore, we had to add the % in both Table 2 and Table 3.

Table 2 and 3 , line 139 and 160

3. There is no need for 2 decimal places in the % reporting - Normal (48.06%) could easily be rounded.

We agree with this comment. Therefore, we had Corrected

Table 3, line 160

4. Would be nice if Table 1 is on one page

Corrected accordingly

line 139 (it became Table 1 after we added Table 1)

5.  Was looking for a bit more detail/discussion around the number of women with previously diagnosed GDM - especially with those having >1 pregnancy and those now with T2DM.

We regret to inform you that due to missing information in the medical records, we were unable to provide the additional details you requested.

These issues were addressed as study weakness

Line 247

6. Line 200/1 "In this study, macrosomia was the most common neonatal finding related to GDM..." - Table 2 disagrees with 25 macrosomia and 26 congenital heart disease.

We corrected accordingly

Line 223

7. Line 215 (also line 52) , it isn't really between 4.3 and 24.3%. According to the introduction

4.3 and 24.3 refer to different measures. 24.2% had GDM and 4.3 pre-GDM so possible maternal diabetes was 28.5% rather than a range between 4.3 and 24.3% (though line 53 has 24.2).

corrected to (Currently, in Saudi Arabia, the prevalence of maternal diabetes is around 28.5% of all pregnancies [8].)

Line 54

The paragraph (lines 46-56) is one which could be reworded to be clearer.

Edited accordingly

line 46

Lines 114-123 please check the alignment with text with the numbers eg 22% with a family history of diabetes does not align with most.

Corrected accordingly, (Most of the patients had no family history of diabetes, 141 (77.9%). )

Line 117

"The findings of this study are supported by data gathered from other studies of the nation" line 195-6 , nation may be better replaced with Saudia Arabia.

Corrected accordingly  (The findings of this study are supported by data gathered from other studies of Saudi Arabia. )

Line 194

Round 2

Reviewer 1 Report

Comments and Suggestions for Authors

Dear authors,

 I read with great interest the manuscript, which falls within the aim of this Journal. In my honest opinion, the topic is interesting enough to attract the readers’ attention. Nevertheless, authors should clarify some points and improve the discussion, as suggested below. Authors should consider the following recommendations:

 In my opinion you have to improve the paper refering to the the updated knowledge on pcos that is linked with GD (GESTATIONAL DIbetes) by nutraceutical supplementation with inositols as well Alpha-lipoic acid,vitamin d and metformin and how in these pts in suggested to freeze oocyte in case of OHSS risk.

I suggest you to read AND CITE THESE ARTICLES:

 Inositols administration: further insights on their biological role Gynaecology & Obstetrics Italian Journal of NARRATIVE REVIEW

Does Alpha-lipoic acid improve effects on polycystic ovary syndrome?

The role of vitamin D in metabolic and reproductive disturbances of polycystic ovary syndrome: A narrative mini-review

Open vs. closed vitrification system: Which one is safer?

Author Response

Article Maternal Diabetes Mellitus and Neonatal Outcomes in Bisha: A Retrospective Cohort Study.

Dearest

Thanks for the reviewer comments on our article, which will surely improve its quality. Herewith, is our response to you. We are sorry for delaying sending it, this is for more critical review and improvement.

Best,

Abdalla Alshomray

Comment

Response

note

Dear authors, I read with great interest the manuscript, which falls within the aim of this Journal. In my honest opinion, the topic is interesting enough to attract the readers’ attention. Nevertheless, authors should clarify some points and improve the discussion, as suggested below.

Thanks for the encouraging words

In my opinion you have to improve the paper refering to the the updated knowledge on pcos that is linked with GD (GESTATIONAL DIbetes) by nutraceutical supplementation with inositols as well Alpha-lipoic acid,vitamin d and metformin and how in these pts in suggested to freeze oocyte in case of OHSS risk.

Once again, we sincerely appreciate your thoughtful feedback, and we are committed to improving the quality and relevance of our paper based on your suggestions.

Line 245

I suggest you to read AND CITE THESE ARTICLES: Inositols administration: further insights on their biological role Gynaecology & Obstetrics Italian Journal of NARRATIVE REVIEW Does Alpha-lipoic acid improve effects on polycystic ovary syndrome? The role of vitamin D in metabolic and reproductive disturbances of polycystic ovary.

Updated accordingly and we added reference 35

Line 250

Reviewer 3 Report

Comments and Suggestions for Authors

Dear Authors,

I accept your changes and reply, but this is nothing new as I wrote previously.

Author Response

Dear Reviewers,

We, as the authors, would like to express our gratitude for sending us your changes. We have carefully reviewed them and we accept them. However, we would like to bring to your attention that the content you've provided is not new information, as we had already mentioned it in our previous communication.

Thank you once again for your valuable feedback. Please do not hesitate to let us know if there is anything else you would like us to address.

Best regards,
ABDULLAH ALSHOMRANY

Round 3

Reviewer 3 Report

Comments and Suggestions for Authors

I accept your reply